# Prostitution and Deservingness in Times of Pandemic: State (Non) Protection of Sex Workers in Spain

**Estefanía Acién González** *  **and Ángeles Arjona Garrido** *

Laboratory of Social and Cultural Anthropology (HUM-472), University of Almería, s/n, La Cañada,
04120 Almería, Spain
* Correspondence: eacien@ual.es (E.A.G.); arjona@ual.es (Á.A.G.)

**Abstract:** During the COVID-19 health crisis, the Spanish Government launched a series of urgent measures to protect the population from its economic effects. At first, it seemed that sex workers would have access to this protection, given that, technically, their access to the star measure, the IMV (anagram in Spanish for Ingreso Mínimo Vital) (minimum living income), was explicitly expressed. However, in the end, this group was excluded as the final text specified that only those deemed to be victims of gender violence, sexual exploitation, or trafficking could access said measure. We propose to study the usefulness of the concept of deservingness of social benefits to explain this lack of protection in a framework that takes into account political power contexts, the empirical observations of sex workers on their level of access to the IMV, and an exploration of its association with the theoretical construct of deservingness. Through a revision of secondary sources, interviews with key informants, and applying discourse analysis, we found these connections and the evident exclusion of sex workers from the social benefit. Likewise, we found that social stigma and moral and ideological judgments are behind this undeservingness and confirm a process of "NGOization" of care for this group that implies the depoliticization and professionalization of civil society entities such as NGOs.

**Keywords:** deservingness; social policy; sex work; social exclusion; stigma; COVID-19

## 1. Introduction

This research arises from an interest in establishing emerging hypotheses to explain why those engaged in sex work are not a group of political concern. During the preceding year, characterized by the profound health and social crisis caused by COVID-19, Spanish policymakers have been concerned about implementing social rescue economic policies. Among its policies, the IMV (Minimum Living Income) is the most significant measure to support the most vulnerable population, including victims of gender violence, sexual exploitation, and trafficking. Moreover, the policy initially mentioned *women in contexts of prostitution*, whether or not they had been legally identified as victims. This text, however, disappeared from the official documents (but not from political discourse or as symbolic representation), indicating that once more, this sector was forgotten and left unrecognized and without access to any financial assistance.

We suspect that it is related to what is known in political discourse as the social or *whore* stigma, the ideological conflict over what t do with prostitution, and how this manifests itself in the political decisions related to whether or not sex workers are deserving of help from the Welfare State.

For all these reasons, we propose an exploratory analysis to determine the extent to which the classic concept of deservingness can help us explain, even tangentially, the *de facto* exclusion of emergency benefits in times of COVID-19 for sex workers in Spain. To achieve this objective, we posed the following research questions, (1) what has been the political position of the government, (2) what is the level of inclusion (or exclusion) to the IMV for

this group; (3) what is the link to the scientific concept of *deservingness* and, (4) what are the main emerging hypotheses of this analysis that can lead to future research regarding the perpetual exclusion of those who engage in sex work from all types of political recognition and rights.

### 1.1. Deservingness: The Transition from Welfare to Workfare

The hegemonic ideology regarding the concept of *deservingness* represents a moral evaluation used to classify the poor according to the notion of *deservingness* (Katz 2013). While this is not the only meaning of *deservingness*, it is the one that best represents the ideology that the Welfare State uses to justify curtailing or even eliminating benefits in the most extreme cases (Peck 2001). The origins of the term and its relationship with poverty regulation are ancient. The seemingly modern idea that the poor should only be helped if they cannot find gainful employment due to *force majeure* dates back to those debates.

Howe (1990, p. 17) writes, "the distinction between the deserving and the undeserving depends on the ideological veracity that work is the source of all rewards". Similarly, Morell (2002, p. 252)[1] also notes "change in the dominant mentality of the time" and adds that poverty goes from having sacred connotations to being criminalized. In Spain, this change in mentality is not as clear. The classic distinction has been between the shameful, indigent, and the marginal poor (the lazy and prostitutes), thus reserving corrective and repressive measures such as prohibiting begging and forcing them to work for this latter group (Maza 1987, pp. 49–50).

Since the twentieth century and even later, after introducing austerity measures during the 2008 crisis, the reciprocity criterion (Van Oorschot 2006) to access social assistance (active job search, acceptance of training courses) became more discernible (Boland and Griffin 2015; Coulter et al. 2017). Other research shows this phenomenon in some social intervention programs of Spanish civil society (Arqueros 2018), in which merit is obtained through the evaluations of volunteers and technicians from beneficiary organizations.

The dismantling of the Welfare State in Spain has been accelerated and extended in the last decade with the application of a regime of austerity with its policies of "structural adjustment" of capital/labor relations. The fiscal stabilization and financial sustainability policies have entailed a deterioration in public services, such as Health and Education (Sarkis 2018; Sarkis and Amarianakis 2020), a decrease in social services, and a reconfiguration of their intervention models and practices.

The State intervenes, therefore, after diagnosing the social needs of an individual and responding by offering social protection developed by professionals who arbitrate from an institutionalized framework, with equipment and resources to prevent, alleviate, or modify processes of *social exclusion*, or promote processes *of inclusion, insertion or social integration.* Ultimately, it is about socially assisting certain groups in this social inclusion/exclusion continuum.

Thus, *social assistance* as an aspect of the Welfare State tends to be based on a needs assessment subordinated to strict limitations based on the availability of resources. These evaluations are ultimately moral in nature and based on (implicit) ideologies of *deservingness*[2] of aid, which, in turn, is based on perceptions of reciprocity.

What happens when the beneficiary of said assistance programs is considered, through their work, not to have paid sufficient taxes? How is that "gift" returned? Social policies and programs of this type apparently "appear to violate deeply held reciprocity norms" (Fong et al. 2005, p. 297), so that when judgments of deservingness come into play, social class or left/right cleavages are displaced, as shown by experimental approximations (Guijarro 2015).

*Deservingness*, therefore, is a social and cultural construct at different levels, local and global. It is used with different connotations within popular ("from below"), conventional (governments, NGOs, etc.), and academic discursive frameworks (Willen 2015). All of the State or third sector assistance openly works with "the action of classifying". Factors such as work ethic, ethnicity, location, social class, gender, or religion are used

to either include or exclude individuals or social groups from accessing full citizenship rights. Whether the classification of "deserving" or "undeserving" is applied depends on where the responsibility for poverty is placed, on the individual, or on society (agency versus structure). Progressive political choices place the responsibility on society, while conservatives place it on the individual (Watkins-Hayes and Kovalsky 2016). Furthermore, engaging in prostitution is not included within the canons of work ethics since it is not considered as such. In which case, how can people who perform a "job" not be recognized?

During the COVID-19 crisis, this logic accompanied the granting of assistance such as the IMV.

Political powers continue to deny those engaged in prostitution their status as workers, the principal requirement for accessing rights or receiving public support. Despite the classic claim of sex workers on this issue (Lamas 2014), prostitution is only recognized as a legitimate work activity in a few countries and is generally an exercise in police, fiscal and health control over the subjects involved (Villacampa and Torres 2013). Few states, such as New Zealand, focus on worker rights (Healy 2008).

In Spain, sex work is not illegal, but the system does not guarantee sex workers any rights (Clua 2015). The demand for the recognition of prostitution as work is fueled by the need to gain access to citizenship rights and overcome the lack of protection in a context where mostly poor and migrant women can only access feminized, precarious, unstable, discredited, and frequently submerged jobs.

Social stigma is closely related to difficulties in recognizing sex work and the social debate associated with it. The traditional stratification of sexual behavior (Rubin 1989) or the essentialist conceptions of female sexuality (Petherson 2000; Garaizábal and Habas 2010) deny women the legitimacy of taking economic advantage of their erotic capital. The *whore* stigma, understood to be the devaluation or pressure exerted on women who engage in prostitution, accompanies some of the social values about how a woman "should be" and "is an effective mechanism to control non-stigmatized women and dissuade them from infringing on existing models" (Juliano 2005, p. 82). Given the above reasoning, we believe that sex workers are excluded and deemed undeserving of institutional, collective, and political support because they are the victims of social and political denial and structural marginalization.

In practice, sex workers have seen their possibilities of economic subsistence diminished (if not altogether extinguished) due to the preventive measures suggested by the health authorities and the closure of brothels. Thus, given these circumstances, we could be facing the allocation (or non-allocation) of social resources according to moral evaluations and not to possession of rights. In this context, we will use the perspective offered by the reflection on merit/deservingness to expound on the criteria and dynamics of inclusion and exclusion to social assistance for those engaged in sex work and what is required in return (abandonment of the activity, demonstration of deservingness). We suspect that social stigma and certain moral and ideological postulates may be behind these decisions.

### 1.2. The IMV in Detail: The Unfulfilled Expectation of Sex Workers

In March 2020, the Spanish coalition government, comprising the PSOE (Spanish Socialist Workers Party)[3] and UP (*Unidas Podemos*) (United We Can)[4] parties, declared a State of Emergency[5]. It also launched a strategy of social policies initiated by inter-ministerial coordination between the Second Vice-presidency of the Government and the Ministry of Social Rights and the 2030 Agenda, under the banner, *Urgent measures in the social and economic sphere to face COVID-19*[6].

The informative political discourse that accompanied the announcement of these policies is aimed at differentiating the strategy of the executive branch to exit the economic crisis (inevitable after mobility restrictions and interpersonal relationships) from the principles of austerity and cuts in social spending applied by previous governments of the PP (Popular Party) as a response to the financial crisis of 2008. For this reason, the coalition government of PSOE and UP highlighted its concern for the most vulnerable subjects.

Proof of this declaration of intent is the motto "No one left behind". Thus, the institutional argumentation incorporates a language sensitive to the plight of the most affected sectors: "the most fragile economies, the smallest companies, the most precarious workers, the poorest families, the self-employed, the unemployed, the most vulnerable people, are the most exposed. The coronavirus does not distinguish between social classes when it threatens our health, but its economic effects do [...] The social articles of the Spanish Constitution [...] are the guarantee that, as a country, when there are difficult times, we behave like a family and no one is left behind" (Ministerio de Derechos Sociales y Agenda 2030 2020).

The list of measures proposed in this context is extensive and is grouped under 14 banners[7]. The only measure which could explicitly refer to persons engaged in prostitution is the allusion to IMV. This measure is promoted as a non-contributory indefinite[8] and stable[9] Social Security benefit, a subjective right that is "received as long as the access requirements are met" representing "the greatest advance in social rights in our country," "a victory for the people" that "makes Spain a benchmark in terms of social justice".[10] The income received, monthly and in 12 payments, would be €469.93 (for an adult living alone) up to a maximum of €1033.85, incorporating a supplement of €103.939 for single-parent families.

The fundamental requirements are intended to be inclusive. Recipients should be at least 23 years old (or 18 if they have dependent minors and do not receive a pension), able to prove one year of legal residence in Spain, and demonstrate economic independence (one year for families and three years for single people). In the case of cohabitants, the requirements included proof that their relationship had existed at least one year before applying, having previously requested benefits to which they may be entitled, being registered as job seekers, or being recipients of housing services, whether social, sanitary, or socio-sanitary in nature. However, additional clarifications regarding access to the IMV state that this benefit will depend exclusively on the applicant's level of income and assets, establishing minimum and maximum levels.

In the clarification of exceptions to the fulfillment of these requirements, mentions of persons who are victims of trafficking, sexual exploitation, or gender violence appear. It states that "they will need to prove this condition through a report issued by the services supporting them or by the public social services" and that in "this case the provision of residential service may be permanent".[11] Thus, the government openly declares itself abolitionist when it defines the group as *victims of sexual exploitation* and places these people at the same level of mistreatment of their rights as victims of trafficking and gender violence. Furthermore, it outlines a particular route of access to the IMV dependent on the NGOs and public social services "that support them". This last detail is especially relevant insofar as it incorporates the need for a social report that is not required of other groups such as domestic workers or other precarious labor sectors to demonstrate that they are worthy of the measure.

Prostitution is recognized as being part of the submerged or informal economy. It is an economic activity with which the government is much concerned since it is a sector that has been highly penalized. Indeed, the situation has become worse as a result of increasingly unsafe living conditions, an escalation in violence, the lack of identification and assistance to victims of trafficking, and the impact of the economic crisis "mainly due to the difficulty of accessing subsidies, the danger of continuing to engage in sex work, and food insecurity" (García and Meneses 2021, p. 258).

According to NGOs such as Oblatas (2021), the pandemic lockdown forced women into apartments and clubs to pay for rooms even though they were not earning an income, incur mounting debts with landlords, pimps, or moneylenders, all while under the constant risk of eviction. In addition, those who stopped their activity on the street could not cover their basic needs or that of their families while at the same time they lost work such as cleaning jobs. Oblatas also warns about the alarming deterioration of mental health among

these sex workers, given the growing expressions of isolation, anxiety, loneliness, lack of protection, apathy, anguish, hopelessness, and stress.

However, as we will see, sex workers are excluded from the IMV unless they are labeled as victims of sexual exploitation and are accompanied by a civil society entity or public social services.

However, shortly after the IMV initiative was launched, the Minister of Equality, Irene Montero Gil, in her public appearances in defense and political dissemination of the measure, adopted the term "women in prostitution contexts" in her speech[12]. By doing so, she made it appear that she was referring to three subsets of women: women in prostitution contexts, victims of sexual exploitation, and victims of trafficking for sexual exploitation. As a result, any woman who engages in sex work can be considered a victim. The idea was to include all three groups as beneficiaries, regardless of whether or not they had been identified as victims of trafficking or exploitation, including immigrants in irregular administrative situations. The point is that, in the end, the reality was very different. Despite this, as we will see in García and Meneses (2021), the announcement generated expectations in the group.

In this sense, it is particularly interesting to see the perceptible contradiction between the symbolic representations of the political discourse issued by the Ministry of Equality and the official documentation about the IMV process.

In the following infographic (Figure 1), from April 2020, propaganda was published by the State administration's social networks about the inclusion of all women *in prostitution contexts* (including irregular migrants), while, as shown in Figure 2, a month later, that mention disappears, leaving only the victims of sexual exploitation and trafficking.

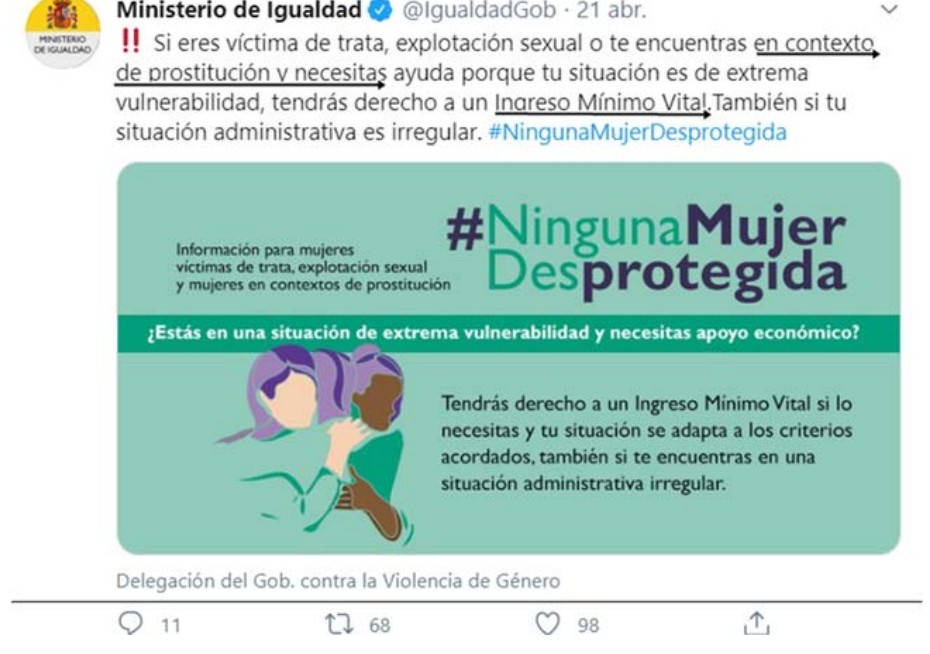

**Figure 1.** Infographic from the Ministry and Equality on access to IMV for women victims of trafficking, sexual exploitation, and women in prostitution contexts. Source: https://twitter.com/ProstitutasSev/status/1377733411792695297?s=20[13] (accessed on 1 June 2021).

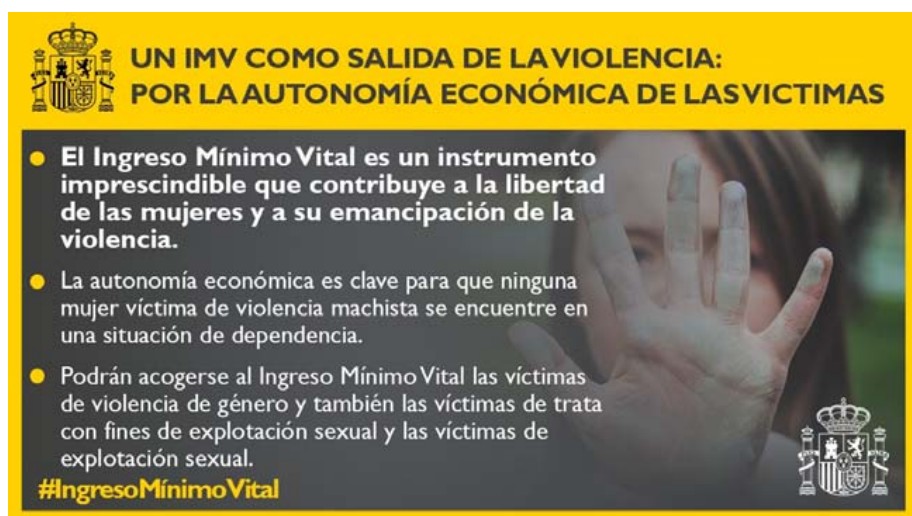

**Figure 2.** Infographic of the Government of Spain published on the IMV's social networks as a "way out of violence". Source: https://twitter.com/Iguald)adGob/status/1266331252015202304/photo/1[14] (accessed on 1 June 2021).

This deletion was likely due to the impossibility of abolitionists to differentiate between *women in prostitution contexts* and *victims of sexual exploitation*, as they considered them all to be included in the latter category. In practice, this meant denying the existence of non-victimized women and, therefore, their exclusion from any state support measure.

Despite this, the Minister of Equality continued to incorporate into her political discourse the alleged effort made by the government for sex workers to access the IMV. Even during the spring of 2021 (a year later), she affirmed that the Executive had "made great strides so that women in prostitution contexts, victims of sexual exploitation and trafficking could access the *Ingreso Mínimo Vital*, even if they are not in a regular administrative situation".[15] This is despite NGOs and activist sex workers warning that the measure had not reached these groups.

As we will see below, the outrage expressed about this issue has been a constant during the most challenging months of the pandemic, both on the part of activist sex workers and civil society. Next, we will address the experience of some sex workers in their (unsuccessful) attempt to access the IMV and, secondly, we will look at aspects related to the relationship of civil society with the processing of benefits for prostitutes and victims of trafficking (including the main criticisms that some entities make about this process).

## 2. Materials and Methods

In order to shed light on these questions, we have worked on a research model inspired by *mixed methods* combining various tools. Regarding secondary sources, on the one hand, the documentary and bibliographic search on *deservingness* and sex work focus on the possible usefulness of the term to explain the level of recognition to access to public resources for sex workers and, on the other, the political discourse of public entities and the access of this group to the IMV.

We also tracked the political discourse on social networks from March to September 2020 (Twitter, Facebook, Instagram, and YouTube), particularly the profiles of public administrations (Government of Spain and Ministry of Equality), government political parties (PSOE and Unidas Podemos), civil society entities and NGOs (Red Cross, Oblatas, Adoratrices, Médicos del Mundo, N.O.M.A.D.A.S.) and sex worker activists, individuals and the collective profiles of CPS (Collective of Prostitutes of Seville), and AFEMTRAS (Asociación Feminista de Trabajadoras Sexuales) (Feminist Association of Sex Workers).

In addition, we have supplemented and triangulated these data with primary sources by conducting semi-structured script-based interviews with four key informants who were selected through intentional sampling. The interviews helped us construct the narrative of the experience of those engaged in prostitution during a crucial period of the health crisis (from March to September 2021) and its relationship with the social benefits proposed by the government. To this end, we recruited two activists sex workers and two members from civil society entities because of their relevant roles in pro-rights activism and prominence within their organizations (CPS and AFEMTRAS, both involved in initiatives to help peers during the lockdown decreed by the government). Moreover, these activists had personal experience of applying for social assistance during the State of Emergency in Spain. The two workers from civil society entities were from APDHA (Asociación Pro Derechos Humanos de Andalucía) (Pro Human Rights Association of Andalusia) and N.O.M.A.D.A.S. (Office of Migration and Attention to Affective Sexual Diversity). These two organizations have close ties with sex worker entities and were primarily engaged in sex work to cover their basic needs during the worst stages of the pandemic.

The procedure used was a recorded videoconference that was then transcribed and analyzed through the thematic and chronological organization of all the information collected to establish the theoretical proposal and the chronicle that constitute the fundamental pillars of the present work.

## 3. Results

In the spring of 2020, the first activist documents appeared regarding sex worker rights denouncing that they could not access the IMV. These documents showed the disappearance of mentions of sex workers from official documents.

A document jointly drawn up by the CPS and APDHA[16] in June 2020 expressed how the Executive's political discourse had fueled the hopes of sex workers of a financial solution (and, more importantly, an acknowledgment of their existence and problems). However, their hopes were later dashed when the facts negated the Executive's words, and they were once more excluded. Both entities requested a meeting with the Ministry of Equality, which was granted and later canceled. As the bureaucratic IMV application processes became increasingly specific, their possibilities of accessing this benefit disappeared, resulting in only one response: prostitutes would have to try to access the IMV based on income levels, similar to any other citizen.

Although they were included in the "Expansion of the Contingency Plan against gender violence in the face of the COVID-19 crisis: additional measures aimed at victims of trafficking, sexual exploitation, and women in prostitution contexts"[17], it was alleged that they did not appear in the technical documents (action guides, proposals to unify criteria and indicators to clarify situations of vulnerability, etc.). According to Sánchez (2020), "none of the 44 pages of the final law mentions the word 'prostitution'. What has happened? How will they access the IMV now? About three months following the start of the State of Emergency, women engaged in prostitution continue to be unable to cover their most basic subsistence needs and suffer mounting debts as they wait for what they were promised".

They were no longer included among the groups that would not have to meet all the requirements, even with the help of a report from social services or civil society. Thus, migrants in an irregular situation, those who were not registered, had not made Social Security contributions because they did not have employment contracts (impossible in the prostitution sector because it is not recognized as work), were excluded. That is to say, practically all sex workers. Once more, they saw how the State "did not recognize them as workers, nor as victims of their lack of protection and condemns them, once again, to live in limbo" (Sánchez 2020).

The feeling of exclusion can be clearly perceived in this fragment of an article published by activists in a well-known political magazine during the pandemic: "A Ministry of Equality that would display its name should recognize whores as subjects, as equals, and

should use the authentic accumulated experience of those who put their bodies into this work every day to consult them on measures that affect them. However, sex workers have once again become the rope in a political tug of war, and politicians are indifferent to how much they affect their lives, even if, amid this fight for a medal, their lives are broken" (García and Sánchez 2020).

### 3.1. Sex Workers vis-á-vis Benefits

Our informants, sex workers and both migrants, devoted a significant amount of time to recounting their experiences regarding the processing of benefits throughout the months of the State of Emergency. One of them, a resident of A Coruña (Galicia) and an independent worker without family responsibilities, did not consider applying for the IMV. She was fully aware she did not meet the required criteria and knew that she was not among the groups recognized as in a deserving special situation. However, she requested information from an NGO about her possibility of accessing benefits. To start, they offered her an information course via WhatsApp on COVID prevention.

> "This made me laugh because the first thing she did was offer me a course and a doctor called me, and, in that video-call, I participated, and there was another girl engaged in prostitution in a shelter, she was in a shelter [...] I said 'What face mask?'. If we want to eat, we have to work. The client does not want to wear a face mask. And she said, 'What are you doing?' Even if you don't negotiate the condom, you have to negotiate the face mask. If they don't want to use the face mask, then you don't kiss [...] And, in the end, she sent me a form that I had to complete, of course! Because that's all subsidized". (Sex Worker 1)

Secondly, the NGO offered to apply to the RISGA (Income for Social Inclusion of Galicia), informing her that the Municipal Social Services of A Coruña would require her to stop sex work to access the benefit. She refused since the benefit would not be more than €300.

> "Because, here in Galicia, there is the RISGA, but to collect it, they demand, if you are engaged in prostitution, they demand you leave prostitution, and it is a miserable income, I don't know if it is €300. If they discovered I was a prostitute, I'd have to pay the money back. When they told me that I had to quit prostitution, I told them that I was not interested. Those €300 would not fix anything. And that I am going to continue in prostitution. And as to whether they could expose me, I am very publicly exposed in a way. It is very easy to discover that I am a prostitute. Because of the ad [...], I am in *Pasión*[18], and I can be found. Then, they were going to find me, regardless. So, I said no because I can't live on €300, 350 or even €400. I have to pay the phone, I have to pay rent, I have to eat [...] What am I supposed to do with €400?". (Sex Worker 1)

Third, the NGO informed her of the possibility of accessing the Food Bank, which she also rejected, although she did agree to be registered in this resource, given that she had always had problems accessing the municipal register.

> "They offered me a basket from the Food Bank, every 15 days. In other words, I could starve to death on the other days. So I said no. [...] I had problems registering and they did help me with that because I could register at the Food Bank of A Coruña. It was managed by a social worker [from a civil society]. And very well managed". (Sex Worker 1)

After attempting to access rent payment assistance and being denied due to lack of an employment contract and registration, the second informant decided not to apply for the IMV due to financial needs derived from family responsibilities both in her country of origin and in Madrid, her usual city of residence. Curiously, she told us how, despite working as a prostitute on the streets, she is the one who financially supports her immediate family.

> "Now my nephew and his mother have arrived. But, the rest are in Ecuador, with my other nephews. My mother is here, but she is a domestic worker. I'm always

helping out my nephews and helping them with food, things [...] And my brother also needs help when something happens. My brother is a supermarket cashier, and there are times when he can't make ends meet. I'm the one with the most stable financial situation. My mother doesn't even earn the minimum wage and my brother earns €900, I think, but he has to pay 800 for rent". (Sex Worker 2)

Based on her experience, she highlights the complexity and slowness of the application procedure.

"I did the Minimum Income thing, which is a mess, it's a bummer, and it took a long time, and then they replied that I needed a letter that said that I gave access to my data. So, time passed, and then they told me it was denied, because, supposedly, there are family ties". (Sex Worker 2)

The difficulties in presenting the documentation and later appealing the denial decision forced her to seek external support. She resorted to the advice of the APDHA and, later, to a lawyer friend to appeal the inadmissibility ruling for living with relatives with incomes. In the description of this long and tortuous process, which had not yet finished at the time of the interview, our protagonist alluded to the disappointment of having believed that she could access the IMV, given the initial announcement of the Ministry of Equality, until she finally understood that, if she did not declare herself a victim of exploitation, she could not do so (except for the income threshold).

"And they had a lot of patience with me, huh? Later on it became a hassle, attach this, fill out that, and I don't know what else. But, I tell you that we added many workers to the IMV application to see if we could get somewhere with it, And no ... no luck! It turns out that, in its day, it was said that women in a prostitution context, well, they said it that way, and it turns out that later, when it was approved, we were no longer *in prostitution contexts* and only the *victims of sexual exploitation* were listed. So, they already took away from how to access that help, right? We had to be declared ourselves to be victims. We are not victims". (Sex Worker 2)

Along this same line, she recounts how some colleagues tried to process their applications through other NGOs, experiencing long waits and little information.

"There are colleagues who are beginning this process: 'Ah! Well, look. The same thing happened to me with the minimum income [...], or, that, it happened to me, they kept me waiting, 'I spent days to get a piece of paper,' 'they did not pick up the phone [...] 'They did not tell me of a specific experience, but they said things like:' I was waiting for many days, 'They did not answer the phone.' That was because they had gone to these organizations to help them". (Sex Worker 2)

They even mentioned the experience of a trans woman who was required by an entity to sign a paper declaring herself a victim of sexual exploitation to be able to proceed with the process with some guarantee of success.

Regarding the group's level of accessibility to the IMV, both informants were outraged and disappointed because they considered themselves exploited by the political discourse and then made invisible and ignored by the State. For them, the key to the malaise lies in the expectations generated by the Ministry of Equality, when it seemed that they were going to be recognized as a vulnerable group, only to discover that they could only access these benefits if they declared themselves to be victims.

"It is a shame. It's an outrage. What they did to generate expectations of that size in women, it's also making fun of the hunger felt by these women. Because it generated such an expectation in the prostitutes, to later leave us with our pants down [...] It's not worth the paper it's written on. That was on April 21. 'Ay! What a nice plan I have, look.' It was an absolute disaster. Society does not see it. It makes me angry when I see Irene Montero, and I read all this in the media, and it burns me. It's a joke". (Sex Worker 1)

> "The current government is entirely abolitionist, and, of course, it was not going to take us into account unless we declare ourselves victims. They no longer count on us, much less on this issue. Although they always try to make us look like real victims, now that we are genuinely in need. That we are victims of hunger, of necessity, we cannot even access a benefit like any other citizen in Spain, of the State, right?". (Sex Worker 2)

This situation only adds to the already tense relationship that thousands of sex workers have with the administration and civil society entities specializing in their care. This tension has already been verified in previous investigations (Acién 2015) and is evident in the speech made by prostitutes, human rights activists, and our key informants' testimonies.

As we have seen, it is common for public services and some entities to urge sex workers to abandon their activity and even declare themselves victims of sexual exploitation or human trafficking to initiate benefit application processes. Moreover, these entities tend to initiate hypervigilance procedures into the private lives of the sex workers, especially concerning the care of their children, specifically, influencing the monitoring of schooling (including through police surveillance), their treatment, their food, who is left in charge when their mothers are working, etc. All this makes women fear disclosing details to the administration and NGOs, more so in the case of migrants in irregular situations, knowing that they are at risk of being deported and seeing their life projects frustrated.

Both of our informants expressed feelings of abandonment, misunderstanding, deception, neglect, or condescension on the part of the administration and the leading civil society entities. They did not hesitate to link it to the power of social stigma and how it manifests itself in public policies and the social intervention system.

> "Yes, because then they can treat us like idiots as if we are obligated to get by on our own. We need to inform ourselves so that they do not deceive us! Watch out! We are a part of a marginalized, forgotten society that no one wants to care for. Not even by giving us the possibility of accessing this help, this benefit, or the Gag Law, which has screwed up our lives. The two things have joined up, the pandemic and the other thing. In fact, in the same spaces where prostitution is talked about, they don't even want to talk to the people involved. I think it is a moral sense that does not allow them to work with us". (Sex Worker 2)

> "We are not recognized as workers. Therefore, we are not listed anywhere; we cannot access absolutely anything. From there, we are already totally excluded. The politicians we have are cowards when addressing a growing reality. Because prostitution is the consequence of the position of women in society. And nobody sees that behind this, there is enormous State negligence. And, of course, also the stigma. The bad woman. They don't want to tackle prostitution because be, Beware! They are not going to contaminate themselves with us. We are polluting agents for their policies, and they are only playing to the gallery. Therein lies the stigma of the dirty woman that pollutes". (Sex Worker 1)

### 3.2. Civil Society Entities and the IMV Process

The intermediation of civil society between vulnerable people and the public administration is a constant in the modern Welfare State, and it is relevant to know which entities can access this role and why. (See Figure 3)

A frequent route is political planning by the government, including key actions and priorities for social intervention and the subsequent calls for proposals and tenders to which entities apply to finance their work. In order to present a financially viable project, it is necessary to design an attractive intervention strategy. For this reason, organizations employ language and narratives along government policy lines that are most likely to be financed. Thus, in the case of the NGOs that intervene in the field of prostitution, the largest and those with an abolitionist discourse are most likely to be considered by the public administration to act as intermediaries with the population they serve. Clemente (2019),

for the case of the intervention against human trafficking, explains very well how "the possibility for organized civil society entities to enter the field of the fight against trafficking, accumulate economic, cultural and social capital, and remain in a central position, and is conditioned, first of all, by sharing this approach to fight against trafficking".[19]

From the moment the information was released that victims of gender violence, women in prostitution contexts, victims of sexual exploitation, and trafficking would be among the groups exempt from meeting the requirements established for accessing the IMV, specialized civil societies were invested in acting as mediators in the bureaucratic process. They did so because they and the public social services would testify to the vulnerability of these people through a report that would serve as evidence.

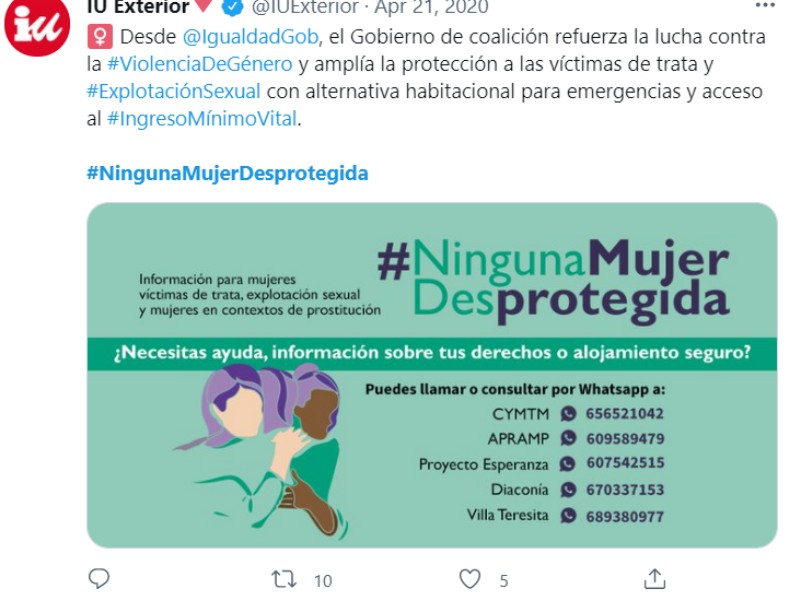

**Figure 3.** IU Twitter publication with an infographic from the Ministry of Equality with contact telephone numbers of civil society (April 2020). Source: https://twitter.com/hashtag/NingunaMujer Desprotegida?src=hashtag_click[20] (accessed on 1 April 2021).

The Expansion of the Contingency Plan[21] proposes an accreditation model for the IMV application for victims of trafficking and sexual exploitation. The model specifies that entities that work in "contact with women in prostitution contexts, directly or through cases derived by FCSE, will present the cases of women they believe show signs of trafficking or sexual exploitation, and of women in situations of isolation and high vulnerability". To this end, "*the report*" indicates the woman's situation and the fulfillment of some requirements that would allow her to access financial aid and/or housing". Furthermore, it states that "for a specialized organization to have the function of accrediting cases for these purposes, it must meet certain minimum requirements", namely, more than 4 years of intervention experience, specialization in work in prostitution contexts, wide territorial presence, and infrastructure, or 24-h telephone service. In this way, the accredited entities "will undertake to monitor the women to whom the assistance is granted, and accompany them in decision-making and accessing resources to attend to their needs".

When it was time to become accredited, issue reports, and carry out the procedure for real people, some of the leading civil society entities in Spain denounced that it was impossible to provide access to the IMV. Thus, for example, at the end of July 2020, *Europa Press* and *Público*[22] published two articles in which the Spanish Red Cross[23] denounced obstacles for victims of trafficking to access the IMV. The entity warned of the non-existence of specific procedures and that they were excluded since most of the possible victims of trafficking were migrants in an irregular administrative situation. "Without a passport or residence permit, it would be impossible to process the application [...]. You can't move

on from the first screen if you don't have an ID".[24] Furthermore, Begoña Vera, a journalist and activist of the Oblate Sisters,[25] explained on social media the problems of application in the context of the Contingency Plan. It recounted how the organizations should have asked that prostitutes and victims of human trafficking be included among the exceptions to the requirements, but, despite allowing victims of trafficking to be included, they found the vast majority of the requests denied.

> "And women victims of trafficking (we have already described those perverse distinctions) have to include Annex 1 and 2 (as those of us who are working on this call them), which accredits their situation of trafficking to guarantee access to the IMV. In other words, we have to have an assessment. Speaking of the IMV, we have had a total of 129 requests [. . . ] 80% of which have no clear response from the administration [. . . ] And the reason for denial in those cases where there has been a response is that the applicant has not paid Social security contributions for more than one year in Spain. Interesting [ironic tone]". (Begoña Vera, videoconference[26])

The same activist decried the lack of foresight of the Ministry of Equality when it required the closure of the brothels without previously consulting with the NGOs specialized in this field and without foreseeing alternatives for the women who would be left without housing or income. On the other hand, it denounces that the existing housing resources are designed for victims of gender violence, according to the Comprehensive Law against Gender Violence,[27] or identified victims of trafficking, but not for women engaged in prostitution. In this regard, the activist makes an interesting reflection on the perverse political use of terms and the contradictions posed by the fact that those who engage in prostitution are spoken of as victims of sexual exploitation but are not considered as such when accessing emergency housing.

> "The Ministry of Equality recommends that Autonomous Communities stop them working in brothels [. . . ] Very good, it shows goodwill. But, ask the entities, the people working on this issue! [. . . ] Because, while it is being done, are homeless women with no access to housing or income offered alternatives? Of course! Now we have the IMV. [. . . ] But what are they going to eat today? Where are they going to sleep? [. . . ] So, I said on the radio: 'Please send the police and social services to knock on the door to see if the women have been left inside.' It happened. [. . . ] And are there resources for gender violence, for example, to provide housing solutions for women in prostitution? Women with children? We call the resources, and then they say: 'No, this is for gender violence'". (Begoña Vera, videoconference)

The same information that this Oblate Sisters activist states was perceived by one of our informants when she requested information from *Proyecto Esperanza* (Hope Project)[28] and *Médicos del Mundo* (Doctors of the World),[29] in this case, due to lack of information and feeling neglected by the administration regarding how to proceed with the application.

> "I contacted Proyecto Esperanza and told them that I was a prostitute and wanted to know how to access the IMV, that I had seen it in the press, they told me that they knew exactly what I knew, that they had no more information. I felt the indignation of the person who spoke to me. All these entities had a meeting with the Ministry of Equality because of their anger about this. And she gave me the phone number of a social worker from *Médicos del Mundo* in Galicia, and she told me the same thing: 'unfortunately, all organizations are the same. We all know what appears in the media and nothing else, we do not know how it will be managed, or by what means, or which ones will see the requirements'". (Sex Worker 1)

We are talking about large organizations such as the Spanish Red Cross, *Hermanas Oblatas*, *Proyecto Esperanza* or *Médicos del Mundo*. If these entities expressed helplessness and

criticism of the management of the IMV for sex workers and victims of trafficking, greater distress could be expected from those with a more discreet weight in the intervention system because of their reduced logistical capacity or less frequent contact with the administration. Thus, for example, APDHA could not participate in the intermediation to process the benefit because it did not have the proper accreditation.

> "These associations had to request it, to be mediators between women ... we were not there as mediators. We were handling a document that explained the IMV for sex workers or women in prostitution contexts, victims of trafficking, and such and that it had to be requested. Then they left us out [...] they were only the victims of trafficking and after a complaint and [...] the usual". (Civil Society 1)

The NOMADAS activist also spoke to us about this issue and added interesting nuances that complicated her advisory work. She pointed out the insecurity generated by not knowing whether or not women would finally be able to access the IMV and, above all, was concerned that sex workers would have to expose or declare themselves victims (especially mothers) and assume the risk of hypervigilance from the State.

> "All this back and forth, 'Ay! Yes. Now they are going to help sex workers', 'then not,' 'Now we are going to remove this law that will go after them' [referring to the article on prostitution in the Organic Law Draft on the Comprehensive Guarantee of Sexual Freedom[30]]. It has generated even more insecurity about these institutions. We never recommend that they say that they are sex workers, much less if they have children: For them, it is crystal clear! The cases that have come to me of sex workers who were being asked about their son and questioning why their son was not in Spain. Because one of them was asking for food aid and she asked me: 'Am I going to involve my son in this? For €50? No, no, no!'". (Civil Society 2)

### 4. Discussion

We have tried to compose a chronicle that illustrates the abandonment by the Welfare State of a group patently affected by the pandemic. Sex workers are extraordinarily present in political discourse, but they are mentioned from moral positions that make them disappear.

There is no political consensus on what to do about prostitution. Throughout the Spanish democracy, the abolitionist positions have been, and are, hegemonic in the left-leaning parties, while the right does not position itself, except exceptionally in a statist regulation, prioritizing the economic, health, and police control of women.

Thus, abolitionism has permeated almost all public policies, which has, as a consequence, the coexistence of the allegation of the activity (it is not illegal, although it is not recognized as work) with the persecution of pimping and trafficking in the Penal Code and a spectrum of laws that end up criminalizing sex workers and their context. The most severe consequences are, on the one hand, the municipal ordinances related to citizen coexistence that persecute and fine women arguing that they are persecuting sexual exploitation or intensive use of public roads (Bodelón and Arce 2018). On the other hand, the Organic Law 4/2015, of March 30, on the Protection of Citizen Security that fines its visibility (Acién and Checa 2020) and, finally, and specifically, for migrant women in an irregular situation, Organic Law 4/2000, of January 11, on rights and freedoms of foreigners in Spain and their social integration. In addition, in autumn 2021, it is expected that the Organic Law for the Comprehensive Guarantee of Sexual Freedom will be debated in the Spanish Congress, the objective of which is to apply criminal measures to those who rent real estate or profit from acts of prostitution. Even with the consent of those who carry them out (Medina 2020).

When the Spanish Government introduced a package of emergency social measures, including the IMV[31] and the Ministry of Equality, embodied in the person of Irene Montero, launched the message with great propaganda that prostitutes were to be included, naming

*women in prostitution contexts* among the groups that would avoid bureaucratic obstacles. This message raised the hopes of migrants in irregular situations, who thought they would finally be recognized as deserving of a social benefit. However, it has become clear how political abolitionism cannot include this grouping in its agenda. Thus, from the moment in which all *victims of sexual exploitation* are considered, this last term swallows up the recognition of the existence of sex workers, their rights, problems, projects, and demands. Indeed, if a collective does not exist, it cannot receive attention.

In this research, a novel theoretical corpus on *deservingness* has been linked to the inclusion of prostitution at the forefront of social interventionism. The results show that these women are outside the protection of the State's social benefits; They are not considered worthy of institutionalized aid. De facto, they have not been able to access this benefit proclaimed as an unprecedented social advance. However, the issue is that, in order to access the IMV, prostitutes had to be made to disappear. They would need to give up considering themselves as workers whose mobility restrictions and interpersonal contact prevented them from maintaining their source of income and instead label themselves as victims without considering themselves as such. In this instance, we are not even talking about reciprocity, as stated by Van Oorschot (2006), but about self-denial, of promising not to do what had been previously performed to guarantee one's livelihood, despite knowing that said commitment could not be fulfilled, since the State will not offer an adequate alternative. Indeed, the only alternatives it offers are minimal monthly benefits or job training that is equally precarious, gendered, and even less profitable.

The results of the fieldwork have validated that the postulate *"deservingness"* can be used to explain the criteria and inclusion/exclusion dynamics of those engaged in prostitution. One in which sex workers should abandon their activity demonstrate they deserve to obtain aid, take courses, and put themselves in the hands of NGOs or public social services.

It is also confirmed that social stigma and moral and ideological judgments are behind this *undeservingness*. In abolitionist political language, this issue is evident, erasing the existence of the entire collective under the umbrella and abstract consideration that all women who engage in prostitution are victims of sexual exploitation and trafficking. In this sense, we must emphasize that, from the abolitionist ideology proclaimed by the current Spanish Executive, it is assumed that a victim is deserving of public support, not because of a question of law and institutional responsibility, but because of a stereotypical vision of *the victim* that presupposes morally flawless behavior. This idea of the victim contrasts with that of a sex worker not coerced by third parties, whose merit of public recognition and support is questioned. From this perspective, if a sex worker falls into financial disgrace, it is her responsibility (she should have dedicated herself to something else and what happens to her is part of the risk assumed), and it is not the State that should take charge of her problems. We do not doubt the power of social stigma here. In this exclusion, we believe it is a manifestation of direct violence, a product of its structural legitimation and cultural norms.

Therefore, they will be doomed to the request of aid offered solely by NGOs and these entities' selection processes and be doubly excluded. We are witnessing what has become known as "NGOization," which refers to the process by which NGOs professionalize to carry out technical interventions, demobilize their clients, and depoliticize their own actions (Kamat 2004).

**Author Contributions:** Conceptualization, Á.A.G.; methodology, E.A.G.; software, E.A.G.; validation, E.A.G. and Á.A.G.; formal analysis, Estefanía Acién González and Á.A.G.; investigation, E.A.G.and Á.A.G.; resources, E.A.G. and Á.A.G.; data curation, E.A.G.; writing—original draft preparation, E.A.G. and Á.A.G.; writing—review and editing, E.A.G. and Á.A.G.; visualization, E.A.G. and Á.A.G.; supervision, E.A.G. and Á.A.G.; project administration, E.A.G. and Á.A.G.; funding acquisition, E.A.G. and Á.A.G. All authors have read and agreed to the published version of the manuscript.

**Funding:** This research was funded by the Laboratory of Social and Cultural Anthropology (HUM-472), University of Almería.

**Institutional Review Board Statement:** The study was conducted in accordance with the Declaration of Helsinki, and approved by the Institutional Review Board (or Ethics Committee) of University of Almería (protocol code UALBIO2022/020) for studies involving humans.

**Informed Consent Statement:** Informed consent was obtained from all subjects involved in the study.

**Conflicts of Interest:** The authors declare no conflict of interest.

## Notes

1. In 12th-century Europe, the eligibility of the poor to receive alms was already based on an assessment of the degree of need and moral criteria (such as being a good Christian and not engaging in immoral occupations such as prostitution) (Tierney 1959). Subsequent research maintains this definition, it is the persistent attempt to classify people according to their deservingness, Katz (2013, p. 1).

2. The term *deservingness* is used mainly in the social construction and regulation of poverty in the Anglo-Saxon world (Katz 2013). In Spain it is not used in the media, or in political discourse, or in the social sciences, unless it is to translate the English term. The Third Sector in Spain uses the term *vulnerable* to refer to the poor who could be considered *deserving*. Although it is not an equivalent term, nor does it appear to share the ideology of *deserving*, in general, intervention actions show that there are implicit ideas of *deservingness* (Arqueros 2018).

3. Spanish Socialist Workers Party: https://www.psoe.es/ (accessed on 1 April 2020)

4. Acronyms for United We Can, an electoral coalition registered in March 2019 and made up of the *Podemos* (https://podemos.info/; accessed on 1 April 2021) and *Izquierda Unida* (https://izquierdaunida.org/; accessed on 1 April 2021) parties on the occasion of the call for general elections in Spain in that year.

5. The Fourth Article, Section b), of the Organic Law 4/1981, of June 1st, of the States of Alarm, Emergency, and Siege, empowers the Government to, in the exercise of the powers attributed to it in Article 116.2 of the Constitution, to declare a State of Emergency, in all or part of the national territory, when health crises occur that involve serious changes to normality (Royal Decree 463/2020, of March 14, by which the State of Emergency is declared for management of the health crisis situation caused by COVID-19, see: https://www.boe.es/buscar/doc.php?id=BOE-A-2020-3692 (accessed on 1 April 2021).

6. https://www.mscbs.gob.es/ssi/covid19/guia.htm (accessed on 1 April 2021).

7. The *Ingreso Mínimo Vital*, Evictions, Rents and Housing: tenants and Landlords, Housing: Mortgages, Labor rights and measures for workers, Protection of workers in vulnerable situations, Self-employed, Vulnerable consumers and families, Small and medium-sized companies, Women, sons and daughters victims of gender violence, Universities—contracts, teachers, academics, assistants, visitors and general staff, Boys and girls, Animals, Autonomous Communities Care funds. and Plan for the transition towards a new normality.

8. Anticipating that it is charged while the situation of lack of income lasts, under the periodic control of Social Security.

9. It is not, therefore, a temporary aid given during the health crisis, but is advertised as the "ultimate safety net for the entire population" for those who lack sufficient income to "reduce the currently high levels of income inequality existing in our society".

10. https://www.mscbs.gob.es/ssi/covid19/ingresoMinVital/home.htm (accessed on 1 April 2021).

11. See note 10.

12. See, for example: https://www.abc.es/sociedad/abci-victimas-explotacion-sexual-cobraran-ingreso-minimo-vital-durante-estado-alarma-202004210932_noticia.html (accessed on 1 April 2021).

13. Translation of Image: Information for women victims of trafficking, sexual exploitation, and women in contexts of prostitution. #No Woman Unprotected. Are you in a situation of extreme vulnerability and need economic support? You will have a right to receive the *Ingreso Mínimo Vital* if you need it and your situation meets the agreed requirements, including if you are in an "irregular administrative situation". Translation of Tweet: If you are the victim of trafficking, sexual exploitation or in the context of prostitution and you need help because you are in a situation of extreme vulnerability, you have the right to a *Ingreso Mínimo Vital*. Even if you are in an irregular administrative situation. #No Woman Unprotected.

14. Translation of Infographic: IMV as a way of out of violence: through the economic independence of victims. The IMV is a necessary instrument for freeing women from situations of violence. Economic independence is key so that no woman suffering from gender violence needs to be economically dependent. You can apply for the IMV as a victim of sexual violence or sexual exploitation, or trafficking.

15. Verbatim transcription of a video published by the Ministry of Equality on Twitter on 29 March 2021: https://twitter.com/IreneMontero/status/1376567485110091779 (accessed on 1 April 2021).

16. Accessible in image format at: https://twitter.com/ProstitutasSev/status/1272952449817546753?s=20 (accessed on 1 April 2021).

17    Full document accessible at: https://observatorioviolencia.org/wp-content/uploads/Plan-Vi%C4%9Bctimas-trata_COVID_def initivo.pdf (accessed on 1 April 2021).

18    https://www.pasion.com (accessed on 22 March 2021), contact page.

19    Own translation.

20    Translation of Tweet: From @IgualdadGob, the Coalition Government reinforces the fight against gender violence and broadens its protection to victims of trafficking and sexual exploitation with alternative housing solutions and access to the IMV.

21    Expansion of the Contingency Plan against gender violence in the face of the COVID-19 crisis: additional measures aimed at victims of trafficking, sexual exploitation and women in prostitution contexts https://observatorioviolencia.org/wp-content/upl oads/Plan-Vi%C4%9Bctimas-treats_COVID_definitivo.pdf (accessed on 22 March 2021).

22    https://www.publico.es/sociedad/trata-personas-coronavirus-denuncian-trabas-victimas-trata-pedir-ingreso-minimo-vital.ht ml (accessed on 22 March 2021).

23    https://www2.cruzroja.es/ (accessed on 1 April 2021).

24    See complete article at: https://www.europapress.es/epsocial/igualdad/noticia-victimas-trata-encuentran-dificultades-tramit ar-ingreso-minimo-vital-cruz-roja-20200729154223.html (accessed on 22 March 2021).

25    http://www.hermanasoblatas.org/ (accessed on 1 April 2021).

26    See selected excerpts from Begoña Veras' videoconference in a post by activist Raj Redlich on Twitter: https://twitter.com/RajR edlich/status/1382268485930221573?s=19 (accessed on 1 April 2021).

27    https://www.boe.es/buscar/pdf/2004/BOE-A-2004-21760-consolidado.pdf (accessed on 22 March 2021).

28    https://www.proyectoesperanza.org/ (accessed on 1 April 2021).

29    https://www.medicosdelmundo.org/ (accessed on 1 April 2021).

30    See draft, in process at the date of writing this work on the website of the Ministry of Equality: https://www.igualdad.gob.es/no rmativa/normativa-en-tramitacion/Documents/APLOGILSV2.pdf (accessed on 1 April 2021).

31    In a press release from the Association of Directors and Managers of Social Services dated 11 November 2021, the following data is noted: three out of four IMV applications have been denied (73%). Almost 100,000 are pending resolution. Only 8.0% of the population (799,203 people) living below the poverty line in Spain benefit from the *Ingreso Mínimo Vital*. The average amount of the benefit per beneficiary is 172 euros per month.

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
