# Peer review of "Prostitution and Deservingness in Times of Pandemic: State (Non) Protection of Sex Workers in Spain"

_socsci, doi:10.3390/socsci11050199_

Round 1

Reviewer 1 Report

The authors investigate the lack of state protection of sex workers during the COVID-19 health crisis in Spain while employing the concept of deservingness. 

The paper is interesting and the research novel. The issues and concerns about the protection of sex workers during the COVID-19 pandemic are pertinent and so is the critical analysis about them. The empirical material is rich, although it is probably not particularly extensive. The article also reflects a strong understanding of the (non)regulation of sex work in Spain. 

I do believe that this piece could be an important contribution to the literature. There are, however, a few significant issues that need to be addressed.

In fact, even if the approach to the phenomenon is creative, namely through the use of the concept of deservingness, it seems that this concept is not well developed and linked to the analysis and discussion. The use of deservingness as analytical tool should be further developed firstly in the Introduction, where it is barely mentioned and sometimes only in the margin of the manuscript (see footnote 2). All analysis and discussion should take into account this concept that only reappears in the last section (“Discussion” – p. 15), where it would be appropriate to strengthen the discussion on the “usefulness of the concept of deservingness of aid to explain this lack of protection” of sex workers in Spain (p. 1).

The article could have further elaborated on the existing literature on sex work during the pandemic that uses the same critical lens that the authors.

The authors should also provide more details on the methodology they used: they “carry out a scan of the political discourse on social networks” (p. 6) but provide little detail about it. When was it posted?  In which social networks was it? How was it made? The authors also interviewed four key informants - two activist sex workers and two workers from civil society bodies. What were the selection criteria? What could have favoured or hindered the empirical research? Etc.

Attention needs to be paid to the logical flow and consistency of the overall argument. The paper needs editing, especially its structure and flow, as well as a correction of typos and grammatical errors, some of which make it difficult to understand the text.   

Author Response

  1.  

In this review, new elements have been introduced in the analysis, note 2 has been changed, and the references have been expanded.

Mainly, the analysis based on deserving what it proposes is a study of reciprocity to those who are supposed to "deserve" social aid, with the incorporation of the IMV and the way in which it makes workers invisible, it becomes clear that there is a moral judgment on who distributes.

  1.  

We have introduced updated references on the impact of COVID-19 among those who practice prostitution in Spain.

  1.  

We have contributed a more detailed description of tools and sampling, in terms of social networks consulted and the recruitment of our key informants.

  1.  

We have incorporated a better explanation about the consideration of prostitution as a job and the relation of our hypothesis to social stigma (end of point 1.1.)

  1.  

Those who participated in this study as key informants gave their consent to the recording of interviews by videoconference. His consent was recorded at the beginning of the interview. It is not necessary for this study to comply with an approval process in an ethics code.

6.

Input and improvement of requested references in text and footnotes.

7.

The use of English has been reviewed and improved.

Reviewer 2 Report

Thank you for the opportunity to review this manuscript. I learned much about the situation of sex workers in Spain and the social support system in Spain during the last two years. I have some comments and suggestions that I hope will be useful for the author(s):

  1. Most importantly, I do not see the necessary information about the study having been approved by an ethics committee. Please provide this crucial information.
  2. I find the literature on deservingness weak and convoluted. One possibility for the authors would be to update this section and refer to the more recent, and more thoroughly developed, literature on the “deserving poor.” Alternatively, I suggest the authors focus on morality/stigma related to sex work as this literature is also well developed.
  3. I don't quite agree with this argument that social support is only given to workers and therefore sex workers are excluded. I think the authors should tie their argument to the literature on the "deserving poor" where there are other factors instrumental in addition to whether people work or not. This is where the extensive literature on sex work stigma would also be relevant. Alternatively, the authors must engage with the extensive literature on whether sex work is work. Without this literature, their argument is very weak.  
  4. The purpose of footnote number 2 is unclear to me. Suggest removing.
  5. There are a relatively large number of instances where the writing is unclear to me. I suggest that the authors spend some energy on a serious rewrite with close attention to the meaning they are trying to convey. These are some examples:
    1. “During the COVID-19 crisis, these logics have accompanied the granting of aid such as the IMV.” What are “these logics”?
    2. “Therefore, it intervenes after diagnosing the social needs in which a response is given through social protection developed by professionals who arbitrate from an institutionalized framework, with equipment and resources that facilitate preventing, alleviating, or modifying processes of social exclusion, or promote processes of inclusion, insertion or social integration.” It is unclear what “it” refers to here. The rest of the sentence contain many ideas. While they all may be important, their importance is reduced by mentioning them all in one sentence.
    3. “Ultimately, it is about socially assisting certain groups that may be found in this continuum of social inclusion/exclusion.” Both “it” and “this continuum” are unclear to me.
    4. “In this regard, in this work, we will use the perspective offered by the reflection on merit to elucidate on the criteria and dynamics of inclusion and exclusion to social support for those engaged in prostitution and what is required in return (abandonment of the activity, demonstration of deservingness), suspecting that social stigma and certain moral and ideological postulates may be behind these decisions.” Again, I find the many ideas in this sentence confusing. I suggest that the authors focus on one of these ideas in their manuscript. As it currently stands, it seems to me try to base their arguments on all of the terms in this sentence without clearly defining what they mean by them: “merit”, “inclusion and exclusion”, “deservingness”, “stigma”, “moral” and “ideological”.
  6. There are a number of instances where the referencing used by the authors need to be improved.
    1. “the Minister of Equality, Irene Montero Gil, in her public appearances in defense and political dissemination of the measure, annexed the term women in prostitution contexts to her speech.” Please provide a reference for this.
    2. “prostitutes would have to try to access the IMV based on income levels, like any other citizens.” Reference needed.
    3. “This is despite civil society entities and activist sex workers warning that the measure had not reached these groups.” Please provide a reference for this statement.
    4. Lines 245 through 253. Unclear what is a quote here and needs to be referenced properly.
    5. State "did not recognize them as workers, nor as victims of their lack of protection and condemns them, once again, to live in limbo" (Op. cit ). Not APA format.
  7. “This deletion was likely due to the impossibility of abolitionists to differentiate between women in prostitution contexts and victims of sexual exploitation, as they considered them all to be included in the latter category.” This need further explanation. Who are the “abolitionists”? The policy maker? Lobby group?
  8. It is important to include much more detail about the recruitment and selection process used to obtain the research participants.
  9. It is important to describe the recruitment and selection process used to obtain the research participants.
  10. It is important to describe the data collection process including the consent process. In particular, who did the interviews and what questions/formats were used to obtain the data?
  11. The authors make many assertions that sex workers were targeted for special treatment. Their argument would be strong if they compared sex workers to one of more occupational groups with similar or identical situation. For example, the following sentence imply that sex workers were the only workers excluded. This argument would be persuasive if the authors showed that another group in a similar situation was included. “Thus, migrants in an irregular situation, those who lacked registration, those who had not contributed to Social Security because they did not have employment contracts (impossible in the prostitution sector because they were not recognized as work), were excluded.”
  12. Lines 538 through 547. This is new information (unless I missed it previously). Please move this to earlier section. Reserve the discussion for interpretations of material already presented.

Author Response

  1. In this review the analysis based on deserving what it proposes is a study of reciprocity to those who are supposed to "deserve" social aid, with the incorporation of the IMV and the way in which it makes workers invisible, it becomes clear that there is a moral judgment on who distributes.

  1. We have introduced updated references on the impact of COVID-19 among those who practice prostitution in Spain.

  1. We have contributed a more detailed description of tools and sampling, in terms of social networks consulted and the recruitment of our key informants.

  1. We have incorporated a better explanation about the consideration of prostitution as a job and the relation of our hypothesis to social stigma (end of point 1.1.)

  1. Those who participated in this study as key informants gave their consent to the recording of interviews by videoconference. His consent was recorded at the beginning of the interview. It is not necessary for this study to comply with an approval process in an ethics code.

  1. Input and improvement of requested references in text and footnotes.

  1. The bibliography is provided in APA format, since Social Sciences accepts references in any style, using a consistent format everywhere, as we have seen in the web site.

  1. The use of English has been reviewed and improved.

Round 2

Reviewer 1 Report

The revision of the article “Prostitution and Deservingness in Times of Pandemic: State (Non) Protection of Sex Workers in Spain” leaves no doubt about the scientific merit of the article. The revision of the methodological section, which has relevant information on the research process, as well as the revision of its English, improved the manuscript substantially. I also appreciated that greater attention was paid to the concept of deservingness.

I have no doubt that I want to see this article published. However, I suggest a minor revision concerning the numerous links that are inserted in the footnotes. Especially the links that may be less stable over time should mention the date of their last consultation. In this regard, I would like to point out, for example, that the link in note 15 on page 6 does not seem to be currently active (it would be useful to verify if changes have been made or if the link is incorrectly reported).

In reviewing the text, the track changes favored the identification of the changes introduced by the authors. However, I would like to suggest to the authors  to indicate more clearly in the cover letter the referees' comments to which they refer and the points in the manuscript where they responded to the reviewers' comments.

Reviewer 2 Report

I have now reviewed the revised manuscript. I note that the quality of writing has been substantially improved and for that I am grateful. I also note that many of my original suggestions have not been attended to. Nevertheless I am willing to now approve this manuscript for publication with one further requirement: The authors must include a statement that their study has been approved by the ethics committee at their home institution and they must also include the approval number.

Sincerely

Reviewer